schizophrenia; hypothyroidism; treatment adherence; L-thyroxin; TSH

**Corresponding author:**
Shay Gur;
Email: sgur1234@gmail.com

#Equal contribution.

# Adherence of patients with schizophrenia to hypothyroidism treatment

Shay Gur[1,2#] ![ORCID], Shira Weizman[2,3#], Haggai Hermesh[1,2], Andre Matalon[2,4], Joseph Meyerovitch[2,5,6] and Amir Krivoy[1,2]

[1]Geha Mental Health Center, Petah Tikva, Israel; [2]Faculty of Medicine, Tel-Aviv University, Tel Aviv, Israel; [3]Abarbanel Mental Health Center, Bat Yam, Israel; [4]Dan-Petah Tikva District, Clalit Health Services, Petah Tikva, Israel; [5]The Jesse Z. and Sara Lea Shafer Institute for Endocrinology and Diabetes, National Center for Childhood Diabetes, Schneider Children's Medical Center of Israel, Petah Tikva, Israel and [6]Chief Pediatrician's Office, Community Division, Clalit Health Services, Tel Aviv, Israel

## Abstract

Adherence to prescription medications is critical for both remission from schizophrenia and control of physical comorbidities. While schizophrenia with comorbid hypothyroidism is common, there is little research on adherence to hypothyroidism treatment in this population. The current study used a retrospective, matched case-control design. The cohort included 1,252 patients diagnosed with schizophrenia according to ICD-10 and 3,756 controls matched for gender, age, socioeconomic status and ethnicity without diagnosis of schizophrenia. All data were retrieved from the electronic medical database of a large health maintenance organization. Retrieved data included demographics, thyroid functionality test results and prescribed medications. Measures of adherence to therapy were used for analyses as were data from follow-ups of patients with hypothyroidism. A diagnosis of hypothyroidism was found in 299 patients, 115 of whom were also diagnosed with schizophrenia. The 184 without schizophrenia constituted the control group. No statistically significant differences were found between the two groups regarding prescriptions for L-thyroxin and TSH levels and number of TSH tests. Adherence of patients with schizophrenia to hypothyroidism treatment was found to be as good as that of individuals without a schizophrenia diagnosis.

## Impact statement

Our results show that psychiatric patients, unlike their adherence to psychotropic agents, adhere very well to their physical illnesses medication regime in this study. Caretakers should therefore change their stigmatizing attitude toward patients with schizophrenia with regard to adherence to hypothyroidism treatment and offer them the same level of care as is offered to patients with no mental health issues.

## Introduction

Schizophrenia is a chronic mental disorder with a worldwide prevalence of about 1% (Rössler et al., 2005). Illness course involves recurrent psychotic exacerbations interrupted by periods of remission with "negative" symptoms, and impairment in functionality increases gradually. Psychotic symptoms usually respond to antipsychotic medications, which are also used for long-term maintenance and prevention (Kane and Correll, 2019). Yet, according to research, due to a multitude of factors, 34–81% of patients discontinue their antipsychotic medications with many studies putting the rate around 50% (Yang et al., 2012; El-Mallakh and Findlay, 2015; García et al., 2016; Lafeuille et al., 2016; Bright, 2017; De Las Cuevas et al., 2017; Velligan et al., 2017).

In the general population, non-adherence to prescribed medications is a common obstacle to effective treatment of many chronic physical disorders, with rates of non-adherence reported to be 43–78% (Osterberg and Blaschke, 2005). Reasons for non-adherence like economic difficulties, lack of awareness of the importance of treatment for the illness, cognitive deficits and poor provider–patient relationship are even more common in patients with schizophrenia (Osterberg and Blaschke, 2005). Stigma-motivated underestimation of patients' ability to cooperate is often involved and may lead physicians to assume that deterioration of a chronic physical disorder in a patient with schizophrenia is due to non-adherence to treatment, thus causing the physicians to refrain from searching for non-mental health-related sources of the deterioration.

Some early studies on patients with schizophrenia pointed to a trend toward lower adherence rates to prescribed medications, relative to the general population. These findings, however, were

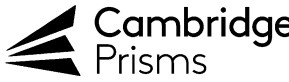



later attributed to the fact that the methods used for estimating adherence were incorrect (Cramer and Rosenheck, 1998). More recent research has shown that patients with schizophrenia have equal or better adherence rates to antidiabetics (Dolder et al., 2003; Simard et al., 2015; Gorczynski et al., 2017), antihypertensives (Dolder et al., 2003; Dolder et al., 2005; Siegel et al., 2007) and antihyperlipidemics (Dolder et al., 2003; Owen-Smith et al., 2016). Additionally, while mental illness, including schizophrenia, has been found to correlate with diagnostic and primary treatment delays in patients with breast cancer, most patients who initiate endocrine therapies adhere to the treatment no less than the general population (Iglay et al., 2017).

Some studies have found a correlation between adherence to antipsychotic medications and adherence to medications prescribed for physical disorders (Hansen et al., 2012; Shafrin et al., 2019). Thus, one may assume that a patient adhering to antipsychotics will also adhere to other medications.

Previous research has shown that there is an increased rate of hypothyroidism in patients with schizophrenia (Melamed et al., 2020). Some studies suggest an association between disorders of the thyroid hormone and mental disorders (Bono et al., 2004; Davis and Tremont, 2007; Radhakrishnan et al., 2013; Remaud et al., 2014; Samuels, 2014). A South Korean study has found that the prevalence of hypothyroidism among schizophrenia patients is 4.9% (Park et al., 2021). A large US study has found that schizophrenia is associated with hypothyroidism (OR 1.88, 95% CI 1.51–2.32) as well as other substantial chronic medical burdens (Carney et al., 2006).

A large-scale Israeli study has found a higher proportion of patients with hypothyroidism among schizophrenia patients than in a control group (2.01% vs. 1.25%, respectively, $p < 0.0001$), after adjusting for age, gender and smoking status. They, furthermore, found a robust independent association between schizophrenia and hypothyroidism (OR 1.62, $p \leq 0.001$) when performing a multivariate logistic regression analysis (Sharif et al., 2018).

Three studies have found that the rate of hypothyroidism seems to rise significantly following a diagnosis of schizophrenia (Telo et al., 2016; Melamed et al., 2020; Launders et al., 2022). These observations indicate that chronic antipsychotic treatment may suppress thyroid functioning (Melamed et al., 2020).

Non-adherence is a known problem in the management of hypothyroidism, with rates of non-adherence reaching 52% (Hepp et al., 2018), but to the best of our knowledge, our current study is the first to compare adherence to treatment and management of hypothyroidism in patients with schizophrenia to that of the general population suffering from this common endocrine disorder.

## Methods

We examined the medical files of 1,252 adult patients with schizophrenia receiving treatment in the regional clinics of a large HMO in Israel, during the years 2005–2013, and of 3,756 control subjects without any mental disorder – matched for age, gender and socioeconomic status – being treated for physical health issues during the same period. These parameters were included in the model as they may affect the dependent variables. Only those under long-term (at least 5 years) care of a healthcare provider, above 18 years of age and residing in the same urban area in central Israel were included. The predefined ratio of patients to controls was 1:3. The relevant data were collected from the electronic medical database and

transferred anonymously to a file, which constituted the data source for the current study. Diagnosis of schizophrenia had been done by senior psychiatrists according to ICD-10 criteria.

Exclusion criteria consisted of treatment with clozapine, which represents treatment-resistant schizophrenia, malignancy and organic brain syndrome. The records of individuals with these medical conditions were not retrieved.

Among the retrieved records, 121 patients were identified with schizophrenia and comorbid hypothyroidism and 190 controls without schizophrenia but with a registered diagnosis of hypothyroidism. Those were included in our study.

Adherence to hypothyroidism treatment was assessed by the following measures:

- Average annual number of prescriptions of L-thyroxine (a replacement therapy for hormone deficiency in hypothyroid state).
- Average annual number of serum thyroid stimulating hormone (TSH) level tests, which is a commonly used test for hypothyroidism.
- Each participant's average TSH level, which is a measure of treatment success.

Due to the retrospective nature of the study, the need for informed consent was waved by the institutional review board of the Geha Mental Health Center in Petah Tikva, Israel. The approval number is 095/2012.

### *Statistical analysis*

Since all the variables are continuous, we used Student's t-test to calculate the differences between groups.

## Results

In our sample, 299 patients had a diagnosis of hypothyroidism corroborated by at least one result of serum TSH levels. One hundred and fifteen had a diagnosis of schizophrenia as well. The 184 with no schizophrenia constituted the control group.

The measures of adherence defined in this study (average annual number of prescriptions of L-thyroxin, average annual number of TSH level tests and average TSH level) are shown in Table 1. No statistically significant differences between the two groups were found in any of the measures.

## Discussion

Several measures used in our study indicated that in patients diagnosed with comorbid schizophrenia and hypothyroidism, the adherence rate to hypothyroidism medications was as good as that of patients with hypothyroidism and no mental disorder. This is in line with previous research on the adherence of patients with schizophrenia and comorbid physical disorders (Dolder et al., 2003; Dolder et al., 2005; Siegel et al., 2007; Simard et al., 2015; Owen-Smith et al., 2016; Gorczynski et al., 2017; Iglay et al., 2017).

Studies consistently demonstrated that patients with schizophrenia have higher rates of physical disorders and a shortened lifespan (Dieset et al., 2016; Hjorthøj et al., 2017). This morbidity and mortality gap has been attributed to a variety of factors, including common biological underpinnings (Crawford et al., 2014), medication effects (Pillinger et al., 2020), suicidality (Hor and Taylor, 2010) and non-adherence (Crawford et al., 2014). As

**Table 1.** Treatment adherence

| | Patients with schizophrenia (N = 115) | Controls (N = 184) | |
|---|---|---|---|
| | Mean ± SD | Mean ± SD | p |
| **Before diagnosis** | | | |
| TSH tests per year | 1.33 ± 1.93 | 1.17 ± 1.46 | 0.707 |
| Mean TSH (mIU/L) | 6.10 ± 6.13 | 7.01 ± 6.71 | 0.098 |
| **After diagnosis** | | | |
| L-thyroxine prescriptions per year | 4.17 ± 3 | 4.78 ± 6.94 | 0.862 |
| TSH tests per year | 1.81 ± 1.26 | 2.02 ± 5.9 | 0.448 |
| First TSH level (mIU/L) | 5.74 ± 7.73 | 7.42 ± 13.58 | 0.848 |
| Mean TSH level (mIU/L) | 5.11 ± 5.57 | 4.79 ± 5.65 | 0.848 |
| Mean TSH level last year (mIU/L) | 4.2 ± 5.37 | 3.37 ± 2.89 | 0.149 |
| Latest TSH level included in the study (mIU/L) | 4.03 ± 3.43 | 3.2 ± 2.79 | 0.072 |

*Note:* Normal range of TSH is 0.4–4.0 mIU/L.

mentioned before, the results of our study indicate that non-adherence is not a significant factor in this context. This finding has implications in two main areas. One is the education of clinicians. Such information could help reduce the stigmatic view that patients with schizophrenia cannot be reliable partners in establishing and implementing a treatment plan (Chaudhry et al., 2010; Briskman et al., 2012; Firth et al., 2016). The second area is resource investment. Economic resources intended to help close the morbidity–mortality gap should be directed to other factors mentioned above rather than to treatment adherence of patients with schizophrenia.

The findings of the current study clearly show that schizophrenic patients adhere to hypothyroid (and probably other) treatments no less than patients without schizophrenia. All patients, regardless of whether diagnosed with schizophrenia or not, should receive the same attention and same treatment from healthcare practitioners. Reducing such stigmatization can and should encourage more equitable and effective healthcare practices for individuals with schizophrenia, both in terms of the attitude from practitioners and the resource investment needed to provide the best available care.

In the case of hypothyroidism specifically, recent research points to the antipsychotic medications playing an important role in the etiology of the disorder (Melamed et al., 2020).

## Limitations

This study has all the limitations inherent to a retrospective study design. In addition, the study group is of relatively small size.

Some studies have found that thyroid function can be affected by early-life psychosocial factors such as childhood trauma. Early life is known to constitute a sensitive period for the long-term effects on the endocrine system, related to the functioning of the hypothalamic–pituitary–thyroid axis (Varese et al., 2012; Machado et al., 2015). However, neither Śmierciak et al. (2022) nor our study could find any reports explaining the dependencies or mechanisms of this relationship.

Additionally, the present study showed no evidence of a relationship between sociodemographic and cultural factors and adherence to treatment, since both the study and the control group were matched for socioeconomic status and cultural background. A future study should investigate the impact of social and cultural backgrounds, healthcare systems and their policies on treatment adherence in patients with schizophrenia. Longitudinal cohort studies are needed to clarify long-term health outcomes associated with treatment adherence or non-adherence in patients with comorbid schizophrenia and how they vary on an international scale. Such studies should also identify the effective strategies to improve adherence in psychiatric patients.

## Conclusion

To the best of our knowledge, this is the first study to address the subject of adherence to hypothyroidism treatment in patients with schizophrenia. Our results boost those of previous studies of adherence in patients with schizophrenia and should help destigmatize the perception of non-adherence as a trait of schizophrenia and the unjustified blaming of these patients as being responsible for the under-treatment they receive.

**Open peer review.** To view the open peer review materials for this article, please visit http://doi.org/10.1017/gmh.2023.86.

**Data availability statement.** Due to the confidential nature of the data, it is only available on reasonable request from the corresponding author.

**Author contribution.** Conceptualization: S.G., S.W., A.K.; Data curation: S.G., S.W.; Formal analysis: A.K.; Funding acquisition: S.G.; Investigation: S.W., H.H., A.M., J.M.; Methodology: S.G., S.W., A.K.; Project administration: S.G.; Resources: S.G.; Supervision: A.K., J.M.; Writing – original draft: S.G., S.W.; Writing – review and editing: all authors.

**Financial support.** This study was supported by a specific grant to SG from the Clalit Health Services, Israel.

**Competing interest.** The authors declare no competing interests exist.

**Ethics standard.** This study was approved by the Institutional Ethics Review Board of the Geha Mental Health Center, Petah Tikva, Israel. The approval number is 095/2012. Due to the retrospective nature of the study, the need for informed consent was waved.

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
