## [Reviewer Report]

To

Prof. Gary Belkin

Editor-In-Chief

Cambridge Prisms: Global Mental Health

Dear Prof. Belkin,

We hereby re-submit our manuscript entitled: “Adherence of patients with schizophrenia to hypothyroidism treatment” for publication in your journal. This is an original study that used an electronic medical database to compare adherence to hypothyroidism treatment of patients with schizophrenia to the adherence to such treatment of patients without schizophrenia. 

The results indicate that, contrary to often held opinions, patients with schizophrenia adhere to hypothyroidism treatment no less than patients without schizophrenia.

All the authors of this article had access to all study data, are responsible for all contents of the article, and had authority over manuscript preparation and the decision to submit the manuscript for publication. All listed authors gave their approval to the submission of the manuscript to this journal. The authors know of no other published, submitted or proposed papers reporting the same or overlapping data.

None of the authors reports any financial or other conflict of interest with regard to this study. 

We hope that you will find this article suitable for publication in your journal.

Sincerely,

Dr. Shay Gur MD

On behalf of the authors

---

## [Reviewer Report]

This is an important and unbiased retrospective study that interrogated a large health system database to identify two groups of patients with hypothyroidism: those with and without a comorbid diagnosis of schizophrenia. The data show that the patients with schizophrenia did not differ from those without schizophrenia in terms of adherence to treatment and treatment monitoring. The data have important public health implications for the management of schizophrenia, encouraging physicians to address this frequent comorbid condition. I am certain the investigators will explore possible psychiatric implications of poor adherence to maintenance of a euthyroid state in what may be a minority of schizophrenia patients with a comorbid diagnosis of hypothyroidism.

The authors may wish to address the discrepancy between the 121 patients with schizophrenia and coexisting hypothyroidism and the 190 patients with a diagnosis of hypothyroidism presented in the Methods, which conflicts with the number of patients presented in the Abstract and analyzed in the Results.

---

## [Reviewer Report]

Dear all,

Thank you for the opportunity to review this manuscript.

This is a relevant paper examining adherence to hypothyroidism treatment in patients with comorbid schizophrenia, comparing them to a control group without schizophrenia. The study shows strengths when addressing and acknowledging the important topic of the stigma associated with clinicians in the field regarding patients with schizophrenia and their adherence to treatment. Another strength of the paper is the objective measures it employs providing a comprehensive view of treatment adherence. The study makes a clear and important conclusion that adherence to medication of patients with schizophrenia was “found to be as good as that of individuals without a schizophrenia diagnosis.” The study’s results are direct and useful for healthcare providers.

However, I believe the study can greatly benefit by further showing how the study’s results specifically fit within the context of global research and how they can inform healthcare practices. The manuscript could include the following elements:

Provide a more extensive review of the literature on treatment adherence in patients with comorbid schizophrenia and hypothyroidism, clearly including and depicting an array of literature around the world. Discussing findings from multiple countries to establish the global context. Further development of this point, the study could include statistics on the global prevalence of schizophrenia and hypothyroidism, and compare these rates across countries to underline any potential variations in morbidity and treatment adherence. Additionally, the study can include sociodemographic and cultural factors, and pinpoint how these factors might vary. To make the study fit the global health context the study can investigate the impact of different healthcare systems and their policies to treatment adherence in patients with schizophrenia. Compare and contrast various countries in the context of effective strategies to improve adherence in psychiatric patients. I also believe the study would benefit significantly from discussing the long-term health outcomes associated with treatment adherence or non-adherence in patients with comorbid schizophrenia, and how they vary on an international scale.

As mentioned above, the study shows strengths when giving light to the important topic of the stigma associated with clinicians in the field regarding patients with schizophrenia and their adherence to treatment. This stigma can influence the way clinicians interact with and treat these patients. Using the evidence provided, the study should clearly depict how reducing such stigmatization can and should encourage more equitable and effective healthcare practices for individuals with schizophrenia.

On a different point, the study provides a simple overview of how the participants were included but lacks specific details about how the inclusion criteria was confirmed, and the process of selecting participants. Specifically, there is little clarity as to how the control group was selected. The study mentions that the control group was matched to the schizophrenia group within the factors of age, gender, and socioeconomic status, but it is unclear how this process was conducted. Furthermore, the study does not mention whether or not there was an exclusion criteria used, if so it should be noted.

Lastly, pertaining to the statistical analysis and results, I think data validation is critical, and the study should mention the steps taken to validate the accuracy of the diagnoses, medication records for all participants, as well as Student’s t-test used to calculate the differences between the groups.

Overall, I believe that the authors did a good job with the paper, talking about an important topic within the world of schizophrenia.

---

## [Reviewer Report]

Nov. 5, 23

To

Dr Franco Mascayano

Handling Editor

Cambridge Prisms: Global Mental Health

Dear Dr Mascayano,

We hereby submit our revised manuscript entitled: “Adherence of patients with schizophrenia to hypothyroidism treatment”. The manuscript has been revised according to all your and the reviewers’ comments.

Our detailed responses follow below.

We hope that you will now find the manuscript suitable for publication.

Sincerely,

Dr. Shay Gur MD

On behalf of the authors

Authors’ Detailed Responses

Handling Editor: Mascayano, Franco

Please also ensure your manuscript complies with the following formatting points (a copy of our author guidelines is included for reference):

Comments

- Please include the abstract in the main text document.

Response: Done

- Please include an Impact Statement below the abstract (max. 300 words). This must not be a repetition of the abstract but a plain worded summary of the wider impact of the article. 

Response: Done

- Submission of graphical abstracts is encouraged for all articles to help promote their impact online. A Graphical Abstract is a single image that summarises the main findings of a paper, allowing readers to quickly gain an overview and understanding of your work. Ideally, the graphical abstract should be created independently of the figures already in the paper, but it could include a (simplified version of) an existing figure or a combination thereof. Graphical abstracts should not be too text-heavy in order to be easily viewable at thumbnail size. If you do not wish to include a graphical abstract please let me know. 

Response: This is a case-control study which will not benefit from a graphical abstract.

- Please ensure references are correctly formatted. In text citations should follow the author and year style. When an article cited has three or more authors the style ‘Smith et al. 2013’ should be used on all occasions. At the end of the article, references should first be listed alphabetically, with a full title of each article, and the first and last pages. Journal titles should be given in full.

Response: Done

- Statements of the following are required in the main text document at the end of all articles: ‘Author Contribution Statement’, ‘Financial Support’, ‘Conflict of Interest Statement’, ‘Ethics statement’ (if appropriate), ‘Data Availability Statement’. Please see the author guidelines for further information. 

Response: Done

- Please submit figures as separate files and please ensure all files are submitted in an editable electronic format.

Response: There are no figures in this manuscript.

Reviewer(s)' Comments to Author:

Reviewer: 1

Comments to the Author

This is an important and unbiased retrospective study that interrogated a large health system database to identify two groups of patients with hypothyroidism: those with and without a comorbid diagnosis of schizophrenia. The data show that the patients with schizophrenia did not differ from those without schizophrenia in terms of adherence to treatment and treatment monitoring. The data have important public health implications for the management of schizophrenia, encouraging physicians to address this frequent comorbid condition. I am certain the investigators will explore possible psychiatric implications of poor adherence to maintenance of a euthyroid state in what may be a minority of schizophrenia patients with a comorbid diagnosis of hypothyroidism.

Response: We thank the reviewer for his or her positive comments.

The authors may wish to address the discrepancy between the 121 patients with schizophrenia and coexisting hypothyroidism and the 190 patients with a diagnosis of hypothyroidism presented in the Methods, which conflicts with the number of patients presented in the Abstract and analyzed in the Results. 

Response: We thank the reviewer for his or her attention. The discrepancy has now been corrected. 

Reviewer: 2

Comments to the Author

Dear all,

Thank you for the opportunity to review this manuscript.

This is a relevant paper examining adherence to hypothyroidism treatment in patients with comorbid schizophrenia, comparing them to a control group without schizophrenia. The study shows strengths when addressing and acknowledging the important topic of the stigma associated with clinicians in the field regarding patients with schizophrenia and their adherence to treatment. Another strength of the paper is the objective measures it employs providing a comprehensive view of treatment adherence. The study makes a clear and important conclusion that adherence to medication of patients with schizophrenia was “found to be as good as that of individuals without a schizophrenia diagnosis.” The study’s results are direct and useful for healthcare providers.

Response: We thank the reviewer for the positive and encouraging comments.

However, I believe the study can greatly benefit by further showing how the study’s results specifically fit within the context of global research and how they can inform healthcare practices. The manuscript could include the following elements:

Provide a more extensive review of the literature on treatment adherence in patients with comorbid schizophrenia and hypothyroidism, clearly including and depicting an array of literature around the world. 

Response: Unfortunately, there is no literature on treatment adherence in patients with comorbid schizophrenia and hypothyroidism. As indicated in the manuscript, our study seems to be the first one investigating this aspect. 

Discussing findings from multiple countries to establish the global context. Further development of this point, the study could include statistics on the global prevalence of schizophrenia and hypothyroidism and compare these rates across countries to underline any potential variations in morbidity and treatment adherence. 

Response: The manuscript now includes descriptions of findings world-wide on the various aspects of the association between schizophrenia and hypothyroidism. They were added just before the closing paragraph of the Introduction.

Additionally, the study can include sociodemographic and cultural factors, and pinpoint how these factors might vary. To make the study fit the global health context the study can investigate the impact of different healthcare systems and their policies to treatment adherence in patients with schizophrenia. Compare and contrast various countries in the context of effective strategies to improve adherence in psychiatric patients. 

Response: A section on these topics was added to the Limitations section of the Discussion.

I also believe the study would benefit significantly from discussing the long-term health outcomes associated with treatment adherence or non-adherence in patients with comorbid schizophrenia, and how they vary on an international scale.

Response: This too was added to the Limitations section of the Discussion.

As mentioned above, the study shows strengths when giving light to the important topic of the stigma associated with clinicians in the field regarding patients with schizophrenia and their adherence to treatment. This stigma can influence the way clinicians interact with and treat these patients. Using the evidence provided, the study should clearly depict how reducing such stigmatization can and should encourage more equitable and effective healthcare practices for individuals with schizophrenia.

Response: The reviewer’s suggestion has been incorporated into the manuscript at the end of the one before last paragraph of the Discussion section.

On a different point, the study provides a simple overview of how the participants were included but lacks specific details about how the inclusion criteria was confirmed, and the process of selecting participants. Specifically, there is little clarity as to how the control group was selected. The study mentions that the control group was matched to the schizophrenia group within the factors of age, gender, and socioeconomic status, but it is unclear how this process was conducted. Furthermore, the study does not mention whether or not there was an exclusion criteria used, if so it should be noted.

Response: In the Methods section there is now a more detailed description of the selection of both, the study group and the controls, as well as exclusion criteria. 

Lastly, pertaining to the statistical analysis and results, I think data validation is critical, and the study should mention the steps taken to validate the accuracy of the diagnoses, medication records for all participants, as well as Student’s t-test used to calculate the differences between the groups.

Response: Since the number of participants in this study was relatively small, data accuracy was validated by examining the data recorded in the medical files for each participant’s last visit. 

Regarding the differences between the groups, Student’s t-test was performed following accuracy validation.

Overall, I believe that the authors did a good job with the paper, talking about an important topic within the world of schizophrenia.

Response: Thank you.

---

## [Reviewer Report]

The authors have thoughtfully considered the critiques of the Reviewers in the preparation of their revision. The revised manuscript is acceptable for publication.